# Effects of CAF-Derived MicroRNA on Tumor Biology and Clinical Applications

**DOI:** 10.3390/cancers13133160

**Published:** 2021-06-24

**Authors:** Xu Wang, Xin Wang, Midie Xu, Weiqi Sheng

**Affiliations:** 1Department of Pathology, Fudan University Shanghai Cancer Center, 270 Dong’an Road, Shanghai 200032, China; xwang15@fudan.edu.cn (X.W.); wangxin_@fudan.edu.cn (X.W.); 2Department of Oncology, Shanghai Medical College, Fudan University, Shanghai 200032, China; 3Institute of Pathology, Fudan University, Shanghai 200032, China

**Keywords:** cancer-associated fibroblasts, microRNA, tumorigenesis, angiogenesis, chemoresistance, biomarker

## Abstract

**Simple Summary:**

Cancer-associated fibroblasts (CAFs), as an important part of the tumor environment (TME), facilitate the progression of tumorigenesis, the development of metastasis, and chemoresistance. MiRNAs—one of the media through which CAFs function—are actively secreted into the TME within exosomes and taken up by specific target cells. Aberrantly expressed miRNAs exert tumor-suppressive or oncogenic functions through negatively regulating gene expression by post-transcription modification. In this review, we describe miRNAs that are differentially expressed by NFs and CAFs, summarize the modulating role of CAF-derived miRNAs in fibroblast activation and tumor advance, and, eventually, identify a potential clinical application for CAF-derived miRNAs as diagnostic/prognostic biomarkers and therapeutic targets in several tumors.

**Abstract:**

Cancer-associated fibroblasts (CAFs), prominent cell components of the tumor microenvironment (TME) in most types of solid tumor, play an essential role in tumor cell growth, proliferation, invasion, migration, and chemoresistance. MicroRNAs (miRNAs) are small, non-coding, single-strand RNAs that negatively regulate gene expression by post-transcription modification. Increasing evidence has suggested the dysregulation of miRNAs in CAFs, which facilitates the conversion of normal fibroblasts (NFs) into CAFs, then enhances the tumor-promoting capacity of CAFs. To understand the process of tumor progression, as well as the development of chemoresistance, it is important to explore the regulatory function of CAF-derived miRNAs and the associated molecular mechanisms, which may become potential diagnostic and prognostic biomarkers and targets of anti-tumor therapeutics. In this review, we describe miRNAs that are differentially expressed by NFs and CAFs, summarize the modulating role of CAF-derived miRNAs in fibroblast activation and tumor advance, and eventually identify a potential clinical application for CAF-derived miRNAs as diagnostic/prognostic biomarkers and therapeutic targets in several tumors.

## 1. Tumor Microenvironment and Cancer-Associated Fibroblasts

Tumorigenesis is a complex and dynamic process, containing the following three critical steps: initiation, progression, and metastasis. The tumor microenvironment (TME), primarily composed of blood and lymphatic vessels, adipose cells, immune cells, fibroblasts, and an extracellular matrix (ECM), is vital for tumorigenesis; the physiological state of the TME is closely correlated with each step of tumorigenesis (see Figure 1) [1,2]. The endothelial cells of blood and lymphatic vessels play a key role in tumor development and immune escape, which offer nutrients and oxygen, evacuate metabolic wastes and carbon dioxide for tumor growth and development, and assist in escaping immune surveillance [3]. Adipose cells, as a major energy source, also contribute to the production of circulating estrogen, recruiting immune cells, and supporting angiogenesis [1]. The immune cells, such as lymphocytes, macrophages, and myeloid-derived suppressor cells (MDSCs), are involved in various immune responses orchestrated by the tumor. Macrophages within the TME promote angiogenesis, ECM degradation, and remodeling, and facilitate tumor cell motility, and are described as “obligate partners for tumor-cell migration, invasion and metastasis” [4]. MDSCs are effective inhibitors of the adaptive immune response to tumors and directly promote metastasis. Meanwhile, tumors are always infiltrated by T regulatory cells (Treg cells), which inhibit adaptive and innate immune in response to tumor stress [5]. Dysregulation of ECM molecules in cancer progression mediates mechanotransduction, as well as tumor initiation and migration [6]. Cancer-associated fibroblasts (CAFs) are activated fibroblasts that constitute prominent components of the TME in most types of solid tumors, and have been shown to facilitate tumor progression by supporting the survival, proliferation, and invasion of tumor cells, promoting angiogenesis, remolding the ECM, and mediating immunosuppression [6].

Fibroblasts are derived from primitive mesenchyme, having an elongated, spindle-like morphology. Normal fibroblasts have a bidirectional effect on tumors: in the early stage of tumorigenesis, fibroblasts maintain the structural integrity of most epithelium tissues against the progression of a tumor; however, as the malignancy advances, fibroblasts are activated to promote tumor development, which are referred to as cancer-associated fibroblasts (CAFs). CAFs are generally identified by the expression of alpha-smooth muscle actin (α-SMA) and fibroblast activation protein (FAP) [7,8]. Several other markers can also be used to identify CAFs due to their heterogeneity, including vimentin, platelet-derived growth factor receptor alpha (PDGFR-α), platelet-derived growth factor receptor beta (PDGFR-β), and fibroblast specific protein 1 (FSP-1) [9].

CAFs, consisting of a heterogeneous population of mesenchymal cells, exhibit systematic differences in different anatomical sites. Within the same type of tissue, a differential expression of CAF markers defines distinct CAF subsets and determines the diversity of the CAFs, which may be attributed to the originate cell types, adjacent tissues, and occasion of activating [10]. CAFs are derived from the following six potential original cell types [8,11] (Figure 2): normal fibroblasts [12], mesenchymal stem cells (MSCs) [13], epithelial cells [14] and endothelial cells [15], human adipose tissue-derived stem cells (hASCs) [16], senescent fibroblasts [17], and cancer stem cells (CSCs) [18], among which CAFs principally originate from the transformation of normal fibroblasts and MSCs.

## 2. Introduction to miRNAs

MicroRNAs, also known as miRNAs, a major class of small non-coding RNAs, are functional single-strand RNAs, ~22 nucleotides long, which mediate post-transcriptional gene silencing by binding to the 3′-untranslated region or open reading frames of target mRNAs. Aberrantly expressed miRNAs exert tumor-suppressive or oncogenic functions by regulating the expression of mRNAs through different signaling pathways, thereby affecting the progression of tumors. MiRNAs are one of the key regulators between CAFs and tumor cells fulfilling the functions of tumor promotion [19,20], with the potential to serve as biomarkers for tumor diagnosis and targets of anti-tumor therapy.

MicroRNAs processed from the introns of protein-coding host genes, as well as other miRNAs, are transcribed in the nucleus by RNA polymerase II, which are termed as pri-miRNAs. The pri-miRNAs are transformed into pre-miRNAs through nuclear cleavage performed by the Drosha RNase III endonuclease. Then, the pre-miRNAs are actively transported from the nucleus into the cytoplasm by Ran-GTP and the export receptor Exportin-5. After being processed by Dicer, miRNAs are incorporated into an RNA-induced silencing complex (RISC) in a single strand form. An RISC directly binds to the 3′-untranslated region (UTR) of a target mRNA, in order to induce post-transcriptional repression, or targets DNA for transcriptional silencing [21] (see Figure 3). Genetic loss, epigenetic modification, extensive transcriptional inhibition, or defective biogenesis are the main causes of dysregulated expression in mature miRNA, in the case of tumors [22].

A range of techniques, such as real-time PCR [23], reverse transcription PCR (RT-PCR) [24], miRNA array [25], and Biotin miRNA pull-down assay [26] have been utilized in many studies to demonstrate the miRNA expression levels in various tumor types. MiRNAs were previously thought to be unstable molecules but have been recently demonstrated to circulate in a highly stable and cell-free form in bodily fluids. Circulating miRNAs can be significantly altered in a wide range of pathological conditions, including cancers. The source of such extracellular miRNAs is still not known, but the following three different pathways have been suggested: (1) passive leakage from broken cells; (2) active secretion from microvesicles, including exosomes and shedding vesicles; and (3) active secretion using a microvesicle-free, RNA-binding protein-dependent pathway, such as high-density lipoprotein (HDL) [27]. Exosomes are small secretory vesicles with a diameter of 40–100 nm, which are key determinants promoting cell-to-cell communication [28]. Specific miRNA populations are selected for packaging into vesicles, and actively secreted into the TME with the assistance of particular molecules. Finally, secreted miRNAs packaged in exosomes are delivered into recipient cells, and act similarly to endogenous miRNAs to exert gene silencing. Similarly, HDL can readily associate with exogenous miRNAs and deliver them to recipient cells [27]. In this review, we do not describe the functions of exosomes and RNA-binding protein in detail.

## 3. Functions of CAF-Derived miRNAs

CAFs in different tumor tissues can produce and secrete miRNAs with different expression levels. Aberrantly expressed CAF-derived miRNAs, either up- or down-regulated, appear to have an enormous impact on the activation of CAFs, tumorigenesis, metastasis, angiogenesis, immunosuppression, and chemoresistance (see Figure 4) [6]. Understanding the relationships between miRNA and CAFs helps to explore tumor biology and possible treatment targets.

In the present review, we describe the differential expression of miRNAs in normal fibroblasts (NFs) and CAFs and summarize the modulating role of CAF-derived miRNAs in fibroblast activation and tumor advance. We also identify the potential clinical application for CAF-derived miRNAs as diagnostic and prognostic biomarkers, as well as targets in anti-tumor therapies.

### 3.1. CAF-Derived miRNAs Promoting Activation of CAFs

CAF-derived miRNAs contribute to the transformation of normal fibroblasts (NFs) into CAFs [29], as well as maintain the activated CAF status. Numerous studies have discussed the possibility of anti-tumor therapy targeting CAFs derived from specific tumors. In consideration of the functions that miRNAs perform to promote the activation of CAFs in various tumor types, miRNAs could serve as another potential target for antitumor therapeutics.

CAF-derived miRNAs in breast cancer have been widely researched in recent years, and their expression levels have been evaluated in pairs of primary NFs/CAFs isolated from patients with different breast cancer sub-types. MiR-9, miR-221, and miR-222 derived from CAFs have been found to be up-regulated, while miR-205, miR-200b, and miR-200c were down-regulated [30,31,32,33]. MiR-222 [30] was up-regulated in the CAFs relative to the matched NFs. MiR-222 directly binds to the 3′UTR of Lamin B receptor (LBR) mRNA, and down-regulates the expression of LBR, which is a direct target of miR-222, leading to the transformation of an NF into a CAF-like phenotype. The knockdown of LBR alone is enough to transform NFs into cells that resemble CAFs, providing evidence that the miR-222/LBR axis can be regarded as the independent factor influencing the conversion of NFs. The significantly higher levels of CAF-derived miR-9 observed in triple-negative breast cancer (TNBC) [31] were involved in the acquisition of a CAF phenotype, while the mechanisms by which miR-9 induces NFs to convert into CAFs are still unclear. Meanwhile, overexpressing miR-9 stimulates tumor cell migration by reducing E-cadherin, which has been demonstrated to be a direct target of miR-9. These up-regulated miRNAs reduce the expression of the corresponding signal molecules and facilitate the activation of CAFs. Inhibitors of such miRNAs or analogues of their target molecules will induce a reversion of CAFs, which may provide a novel method to improve prognosis and prolong overall survival but requires further clinical research to be demonstrated. In contrast, miRNAs that are down-regulated in breast cancer inhibit the suppression of target genes and enhance the downstream signal pathways to promote CAF activation. For example, TGF-β1 derived from tumor cells [32] regulates the expression of miR-200s (miR-200b/c) and miR-221 within CAFs. TGF-β1, together with CAF-derived miR-200s (miR-200b/c) and miR-221, constitutes a self-stimulating pattern by targeting DNMT3B, which is necessary for maintaining the activation of CAFs. TGF-β1 down-regulates miR-200s (miR-200b/c) expression. The response to TGF-β1, the expression of DNMT3B, and the autocrine signaling of TGF-β1 are regulated by miR-200s and miR-221, which are vitally important for maintaining the activation of CAFs. Exogenous TGF-β1 activates the miR-200b/c/miR-221/DNMT3B regulatory loop within the CAFs and, in turn, the miR-200b/c/miR-221/DNMT3B feedback loop influences TGF-β1 expression and secretion within CAFs. The inhibition of either exogenous or autocrine TGF-β1 or DNA methylation by DNMT3B will break the positive feedback loop and provide a potential therapeutic opportunity aimed at reversing the activated state of CAFs into a normal, inactivated fibroblastic state to prevent cancer progression, which may provide a potential target for anti-CAF therapeutic strategies. In addition, miR-205 is down-regulated in CAFs [33], which has been associated with higher CD31 levels and a lower overall chance of breast cancer survival. As a direct target of miR-205, YAP1 is essential for CAF activation.

The MiR-21 derived from CAFs has been confirmed to be up-regulated in several tumors, such as lung adenocarcinoma and pancreatic cancer, and demonstrated to be involved in the conversion of CAFs. The up-regulation of miR-21 [34] in CAFs in the invasive region of lung adenocarcinoma induces a CAF-like phenotype in lung fibroblasts, which is related to the up-regulation of calumenin. In pancreatic ductal adenocarcinoma (PDAC) [35], a high expression of miR-21 mediates the activation of CAFs by targeting PDCD4 and promotes the desmoplasia of PDAC. Meanwhile, the high expression of miR-21 in CAFs elevates the MMP-3, MMP-9, PDGF, and CCL-7 expression and positively regulates the tumor-promoting role of CAFs. In addition, miR-21 [36] is involved in the metabolic alteration of CAFs and affects the development of tumor cells. Examining the glycolytic metabolism in CAFs indicates that miR-21 mediates the reverse Warburg effect phenomenon in CAFs and is involved in CAF–cancer cell metabolic coupling, consequently promoting tumor progression. Although the precise mechanism of the miR-21-mediated activation of CAFs is not clear, miR-21 inhibitors can prevent the transformation of CAFs from the source, consequently suppressing the tumor-promoting effects of CAFs.

In addition to the above, miRNAs can also serve to limit the transformation of CAFs through the negative feedback loop. In oral squamous cell carcinoma (OSCC) [37], miR-145 levels are elevated in CAFs, compared with NFs, and miR-145 targets multiple components of the TGF-β signaling pathway, and acts in the miRNA-145/TGF-β1 negative feedback loop to dampen the acquisition of myofibroblast traits and suppresses biomarkers of myofibroblast activation, suggesting that miR-145 up-regulation may serve to limit myofibroblast differentiation. 

Summing up the above, a great portion of aberrantly expressed miRNAs in CAFs activate multiple signaling pathways to promote the transition from NFs into CAFs, consequently enhancing the proliferation, invasion, and migration of tumor cells. In contrast, a small portion of miRNAs have been shown to negatively regulate the conversion of NFs into CAFs. The inhibition of positive-regulating miRNAs and the stimulation of negative-regulating miRNAs, as well as the inhibitors or analogues of target molecules, may be efficacious in therapies targeting CAFs.

### 3.2. CAF-Derived miRNAs Promoting Tumor Progression

MiRNAs released into the TME by CAFs can affect various characteristics of the tumor cells. Increasing evidence has claimed that miRNAs are capable of regulating the pro-tumor effects of CAFs, including tumorigenesis, development, epithelial to mesenchymal transition (EMT), and metastasis.

A great portion of miRNAs has been found to be down-regulated in CAFs and are released into the intercellular stroma packaging by exosomes. Down-regulated CAF-derived miRNAs relieve the inhibition of their targets, thus promoting the initiation, survival, proliferation, and worse prognosis of tumors by oncogene activation, cell cycle regulation, decreasing the apoptosis ratio, and promoting the EMT. In triple-negative breast cancer (TNBC) [38], miR-4516 was more strongly down-regulated in CAF-derived exosomes than in NF-derived ones, which was associated with the poor prognosis of TNBC patients. The loss of miR-4516 contributes to the proliferation and malignancy of TNBC cells by relieving the suppression of FOSL1, an oncogenic proliferation-related gene (PRG) targeted by miR-4516. The expression of miR-3188 is reduced in exosomes and their parental CAFs from head and neck cancer (HNC) tissues [39]. The down-regulation of miR-3188 increased the expression levels of its direct target BCL2, an anti-apoptotic regulator in HNC cells, to promote the G1 to S cell cycle transition and colony-formation ability, as well as decreasing the apoptosis ratio in tumor cells. In contrast, overexpressing miR-3188 inhibits the proliferation and promotes the apoptosis of HNC cells by down-regulating BCL2. In oral squamous cell carcinoma (OSCC) [40], the expression of miR-34a-5p in CAF-derived exosomes is significantly reduced and the direct target of miR-34a-5p, AXL, is up-regulated, which mediates the proliferation and motility of OSCC cells by increasing the activation of β-catenin, which could activate SNAIL transcription to promote EMT. The miR-34a-5p/AXL axis promotes EMT and cell invasion through the AKT/GSK-3β/β-catenin/SNAIL signaling pathway. CAF-derived miRNAs, down-regulated in CAFs and exosomes secreted by CAFs, aggravate tumor progression through multiple mechanisms, the overexpression of which can restrain the malignancy advance, thus providing a new direction for chemotherapy.

Down-regulated CAF-derived miRNAs also contribute to the migration of tumor cells. In gastric cancer, miR-214 and miR-139 are significantly down-regulated. CAF-derived low-expressed miR-214 [41] removes the inhibition on FGF9 and EMT of gastric cancer cells, then enhances the capacities of migration and invasion of gastric cancer cells by means of decreasing E-cadherin and increasing N-cadherin and Snail expression. MMP11, as a key regulator of extracellular matrix degradation, is negatively regulated by exosomal miR-139 derived from the CAFs of gastric cancer [42] and contributes to gastric cancer cell migration. In addition, the miR-148b target DNMT1 in endometrial cancer [43] and the miR-15a target PAI-2 in cholangiocarcinoma [44] have similar functions in enhancing metastasis through the induction of EMT. The CAF-derived miRNAs that have been identified to be down-regulated in various types of tumors function as tumor suppressors, inhibiting the metastasis of tumor cells by targeting their downstream genes. Raising the expression of these miRNAs or decreasing the expression of their target genes will be conducive to preventing tumor metastasis.

A fraction of miRNAs are up-regulated in CAFs and released into the intercellular stroma packaging by exosomes. Up-regulated CAF-derived miRNAs enhance the metastasis capacity of tumors by activating multiple signaling pathways. For example, CAF-derived exosomes showed the higher expression of miR-17-5p in colorectal cancer (CRC) tissues [45], which led to the down-regulation of its target, RUNX3. RUNX3 then interacts with MYC, and they both bind to the promoter of TGF-β1, thereby activating the TGF-β signaling pathway and contributing to tumor progression. In addition, RUNX3/MYC/TGF-β1 signaling sustains autocrine TGF-β1 to activate CAFs, while the activated CAFs release miR-17-5p to CRC cells, thus forming a positive feedback loop for exacerbating CRC progression. Similarly, miR-382-5p in oral squamous cell carcinoma (OSCC) [46] and the miR-1288 target SCAI in osteosarcoma [47] are also up-regulated to promote the migration of tumor cells. Up-regulated CAF-derived miRNAs are always associated with a positive feedback loop for tumor progression, which may result in a cascade effect that is difficult to revert, such that it is necessary to break out the positive feedback loop at the first step by counteracting these up-regulated miRNAs.

Other CAF-derived miRNAs can play a role in curbing the occurrence and progression of tumors, such as up-regulated miR-34 in gastric cancer [48] and down-regulated miR-320a in endometrial cancer [49] and hepatocellular carcinoma [50]; however, these will not be discussed in detail here.

In summary, CAF-derived miRNAs serve a function of inducing EMT and promoting the initiation, survival, proliferation, invasion, and migration of tumor cells, which has been associated with worse prognoses. The expression levels of miRNAs in the CAFs or exosomes secreted by CAFs could serve as biomarkers to evaluate the potential for tumor progression and metastasis. MiRNA mimics, which compensate the verified down-regulated miRNAs, and miRNA antagonists, which neutralize the proven up-regulated miRNAs, are expected to be efficient in oncotherapy.

### 3.3. CAF-Derived miRNAs Promoting Angiogenesis

In addition to CAFs and tumor cells, aberrantly expressed miRNAs act on endothelial cells to promote angiogenesis. Several CAF-derived miRNAs are down-regulated in specific tumor tissues, and the overexpression of their target genes facilitates tumor angiogenesis. In placental site trophoblastic tumors (PSTTs) [51], one of the most abundant vascular tumors, miR-363 inhibits the expression of EGR1, which is involved in the angiogenesis of PSTT. EGR1 promotes Ang-1 secretion in CAFs, thus promoting the tube formation of human umbilical vein endothelial cells (HUVECs). The inhibition function performed by miR-363 can be recovered by lnc003875, while lnc003875 exerts a negligible effect on miR-363 expression. The lnc003875/miR-363/EGR1/Ang-1 axis in CAFs is crucial for the angiogenesis of PSTT. In breast cancer [33], the miR-205/YAP1 in CAFs displays a concordant effect in VEGF-independent angiogenesis, by targeting IL-11 and IL-15 to activate the STAT3 signaling pathway in endothelial cells, which may provide a possibility for resistance to anti-VEGF therapy.

The neovascularization of solid tumors facilitates the proliferation and migration of tumor cells by providing nutrient flow and a frequently incomplete, fenestrated endothelial barrier between neoplastic cells and circulation. Tumor angiogenesis is essential to provide adequate nutrition for tumorigenesis and tumor progression [52]. Anti-angiogenesis therapy has been proven to be effective in various tumor types, through targeting the cytokine VEGF; however, the promotion function of angiogenesis induced by CAF-derived miRNAs appears to be independent of VEGF, the main target of anti-angiogenesis therapy, which may be why the resistance emerges after tumor anti-angiogenesis therapy. In consideration of the angiogenesis promotion function of low-expression miRNAs, mimics of these miRNAs could play an assisting role in anti-VEGF therapy.

### 3.4. CAF-Derived miRNAs Promoting Immunosuppression

CAFs have been found to enhance the recruitment, differentiation, and survival of T regulatory cells (Treg cells), which are generally identified through the expression of CD25 and the transcription factor FOXP3, contributing to the generation and maintenance of an immunosuppressive microenvironment [6]. According to the expression levels of CD29, FAP, FSP1, and SMA, CAFS in High-grade serous ovarian cancers (HGSOCs) are distinguished into the following four sub-populations: CAF-S1, S2, S3, and S4 [53]. HGSOCs are mainly enriched in activated CAF-S1 and CAF-S4 subsets, among which the HGSOCs enriched in CAF-S1 cells are particularly highly infiltrated by FOXP3 + T cells. Compared with CAF-S4 cells, miR-141 and miR-200a are down-regulated within CAF-S1 cells, leading to the specific accumulation of CXCL12β, the target of miR-141/200a, in CAF-S1 fibroblasts. Thus, the accumulation of CXCL12β further facilitates the recruitment of CD25+ FOXP3+ Treg cells. Treg cells play a key role in fostering an immunosuppressive microenvironment, and the infiltration of Treg cells predicts a shortened overall survival [54].

In addition, miR-92 has been found to be significantly up-regulated in CAF-derived exosomes in breast cancer. After being transported into breast cells, miR-92 silences its direct target, LATS2, which inhibits the inhibition of YAP1 and promotes the nuclear translocation of YAP1. The occupation of YAP1 in enhancer regions of PD-L1 will increase the transcriptional activity of PD-L1, which promotes T-cell tolerance and host immunity escape [55]. In consideration of the function of CAF-miRNAs to promote immunosuppression, the expression levels of miRNAs may be a prospective biomarker to assess the immune status and predict the efficacy of immunotherapy.

### 3.5. CAF-Derived miRNAs Promoting Chemoresistance

Chemoresistance is a serious obstacle to the treatment and management of various types of tumors. Chemoresistance—both acquired and innate—has been associated to complex multifactorial processes, including decreased intracellular drug concentrations, hidden drug targets, aberrant regulation of cell survival, and crosstalk between the TME and tumor cells [23], among which CAFs play a critical role in the regulation of chemoresistance in various tumor types. Among the various functional components derived from CAFs, miRNAs have been extensively reported to play a critical role in cell–cell communication.

The up-regulation of CAF-derived miRNAs mediates the innate resistance of CAFs to chemotherapeutics and transfers the resistance to tumor cells by regulating the cell cycle and inhibiting tumor cell apoptosis. In HNC [56], the innate chemoresistance to cisplatin is mediated by up-regulated CAF-derived miR-196a, which is transferred to HNC cells and promotes the survival and proliferation of tumor cells. Up-regulated miR-196a directly targets CDKN1B and ING5, which perform different functions in miR-196a-mediated cisplatin resistance. Ectopic CDKN1B expression rescues the miR-196a-mediated G1/S cell cycle transition, while exogenous ING5 expression rescues the apoptosis mediated by inhibited miR-196a. CAF-derived miR-196a mediates cisplatin resistance in HNC cells through CDKN1B and ING5 down-regulation, leading to G1/S cell cycle transition and the inhibition of apoptosis. Similarly, CAF-derived miR-130a mediates the chemoresistance to cisplatin in non-small cell lung cancer (NSCLC) cells [23] and CAF-derived miR-106b promotes the resistance to gemcitabine by targeting TP53INP1 in pancreatic cancer [57]. In a nutshell, CAFs with innate chemoresistance transfer their tolerance into tumor cells by up-regulated miRNAs, which contribute to the G1/S cell cycle transition and the inhibition of apoptosis to promote the survival and proliferation of tumors treated chemotherapeutically.

Dysregulation of CAF-derived miRNAs mediates the acquired chemoresistance of CAFs and transforms the tolerance of tumor cells to chemotherapeutics by regulating the cell cycle and inhibiting ferroptosis and apoptosis. Ferroptosis is a novel form of regulated cell death [25], containing iron-dependent peroxides’ (lipid-ROS) accumulation and leading to lethal cell damage, which has been positively correlated to ALOX15, the direct target of miR-522. In gastric cancer, up-regulated miR-522 derived from CAFs suppresses ferroptosis and promotes the acquired chemoresistance to cisplatin and paclitaxel. Chemotoxicity promotes miR-522’s up-regulation and secretion from CAFs by activating the USP7/hnRNPA1 pathway. Subsequently, miR-522 negatively regulates ALOX15 expression at the post-transcriptional level, thus blocking the accumulation of lipid-ROS, suppressing the ferroptosis induced by chemotherapy, as well as the consequent development of chemoresistance. In CRC [26], up-regulated CAF-derived miR-93-5p induces the radio resistance of CRC cells by targeting FOXA1 through activation of the TGF-β signal pathway. FOXA1 negatively regulates the transcription of TGFB3 and prevents its translocation into the nucleus, due to which down-regulated FOXA1 promotes the transcription of TGFB3 and then activates the TGF-β signal pathway, which leads to the G1/S transition and inhibition of apoptosis and, consequently, induces the radio resistance of tumor cells. In breast cancer [58], down-regulated CAF-derived miR-29b exhibits chemoresistance to paclitaxel by targeting CCL11 and CXCL14, which contributes to decreasing the apoptosis ratio of tumor cells. These studies have described the acquired resistance of CAFs, where miRNAs facilitate resistance by regulating the cell cycle and inhibiting ferroptosis and apoptosis.

In summary, aberrantly regulated CAF-derived miRNAs are transferred into the tumor cells and have a variety of functions in the progression of tumor and chemoresistance. Developing an in-depth knowledge of the functions that CAF-derived miRNAs perform in the chemoresistance of tumor cells is conducive to overcoming the barriers of oncotherapy and improving the prognosis of tumor patients.

## 4. Clinical Applications of CAF-Derived miRNAs in Tumors

As was mentioned above, miRNAs play an outstanding role in the process of tumorigenesis, by regulating the cell cycle, metastasis, angiogenesis, metabolism, and apoptosis. The expression of miRNAs in a tissue- and development-specific manner indicates that miRNAs are promising markers for the early diagnosis and prognosis of tumors [59]. As a modulator that essentially regulates gene expression through the post-transcriptional regulation of mRNA, the dysregulation of miRNAs emerges before the phenotypic changes observed in tumorigenesis, and the differential expression patterns of these miRNAs can be perceived at any stage of the progression of tumorigenesis, allowing us to observe the changes in a real-time, dynamic manner [22]. Therefore, miRNA biomarkers are more sensitive and specific in diagnosis and prognosis than the currently used DNA, protein-coding RNA, or protein biomarkers.

The properties of miRNAs that can be detected in cell-free or exosome forms, in both tumor tissues and the bloodstream, are their major advantages over other carcinogenic biomarkers [6]. MiRNAs, by means of active secretion, apoptosis, or necrosis, directly enter the bloodstream from tumor tissues and are stable and detectable in the peripheral blood. Therefore, changes in the number of miRNAs in circulation can reflect the pathological process of tumors. Several CAF-derived miRNAs have been found to contribute to the pathological mechanism of tumor progression, some of which are regarded as non-invasive diagnostic or prognostic markers; for example, miR-145-5p and miR-191-5p (either individually or in combination) in plasma can help to distinguish breast cancer patients from healthy individuals accurately and, therefore, can be considered as potential biomarkers for breast cancer early screening in the Kazakh population [60]. The levels of seven miRNAs—namely, let-7a, miR-1229, miR-1246, miR-150, miR-21, miR-223, and miR-23a—were significantly higher in the serum exosomes from CRC patients than those of healthy controls, which suggests the utilization of serum exosomal miRNAs for the early detection of primary CRCs [61]. In addition, the loss of miR-4516 has been associated with a poor prognosis of triple-negative breast cancer (TNBC) patients [38]. High miRNA-200a expression in stromal fibroblasts may predict a good prognosis in patients with NSCLC [62].

In conclusion, cancer-related circulating miRNAs or tissue specific miRNAs are stable and detectable in the peripheral circulation, such that they can be utilized as non-invasive biomarkers to identify patients at an early stage, to monitor the cases during therapy, and to estimate the prognosis of patients, which may become a promising direction for further research.

## 5. Conclusions and Future Perspectives

In the past few years, increasing evidence has indicated the importance and participation of CAFs in tumorigenesis, development, invasion, metastasis, immunosuppression, and chemoresistance in various types of tumors. In this context, miRNAs have been demonstrated to play a crucial role in the transformation of NFs into CAFs and in the progression of angiogenesis and chemoresistance. Aberrantly expressed CAF-derived miRNAs contribute to the activation of numerous signaling pathways by directly binding to their targets, thus facilitating the activation of CAFs and the pro-tumor functions of CAFs. In addition, several miRNAs possess the potential for use in screening patients from healthy controls at an early stage, dynamic observations of the progression, and predicting a prognosis.

Although pre-clinical and clinical investigations performed on miRNAs have indicated great promise for the establishment of accurate, non-invasive biomarkers, there are considerable limitations to their clinical utility, due to the influence caused by different factors, such as the diversity of sample types, detection methods, tumor heterogeneity, and the ethnicity of patients. Therefore, studies in large homogeneous populations are required in order to investigate the value of these miRNAs as diagnostic and prognostic biomarkers.

Finally, investigations of the molecular mechanisms by which CAF-derived miRNAs promote the activation of fibroblasts and the progression of tumorigenesis, angiogenesis, immune escape, and chemoresistance indicate the potential of CAF-derived miRNAs as a target in anti-tumor therapies. However, there are some barriers of RNA therapeutics that have yet to be overcome, such as the lack of a reliable administration route and effective carriers, lack of optimal dosage regimes, and insufferable side effects. Although it may be too premature to affirm the possibility of using CAF-derived miRNAs as targets in anti-tumor therapy, exploration in the field of miRNA-based therapeutics may provide fruitful directions for further research.

## Figures and Tables

**Figure 1 cancers-13-03160-f001:**
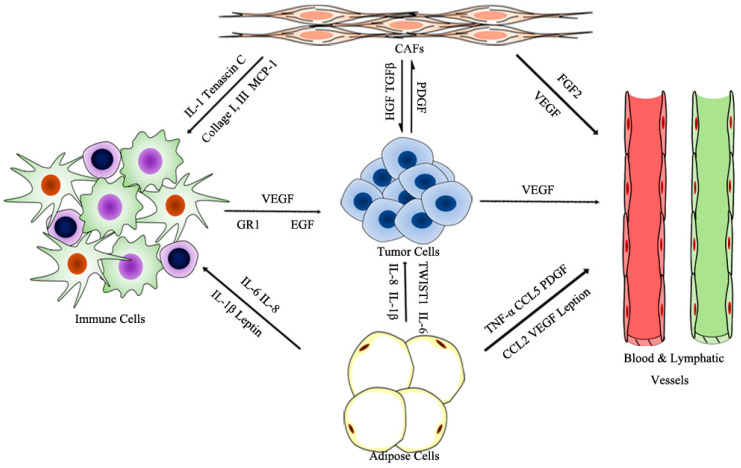
Prominent components of the TME.

**Figure 2 cancers-13-03160-f002:**
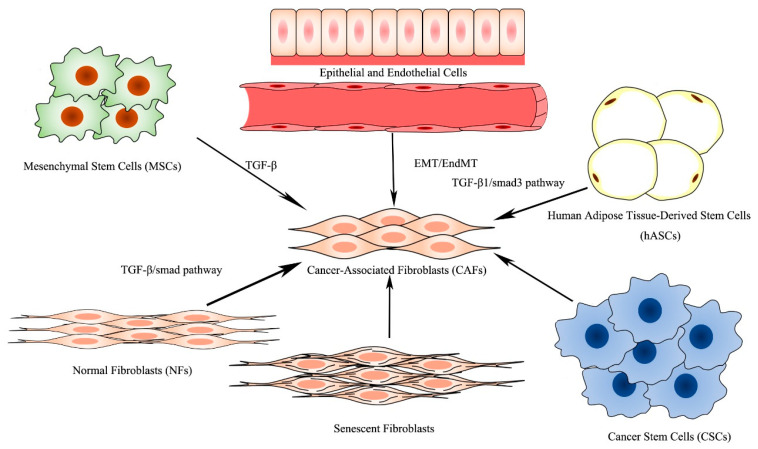
Origin of CAFs.

**Figure 3 cancers-13-03160-f003:**
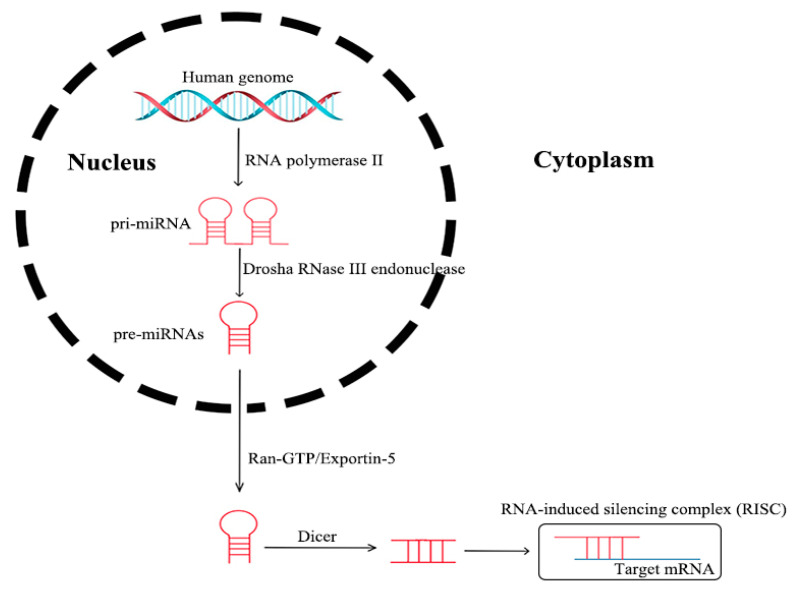
Processing of miRNAs.

**Figure 4 cancers-13-03160-f004:**
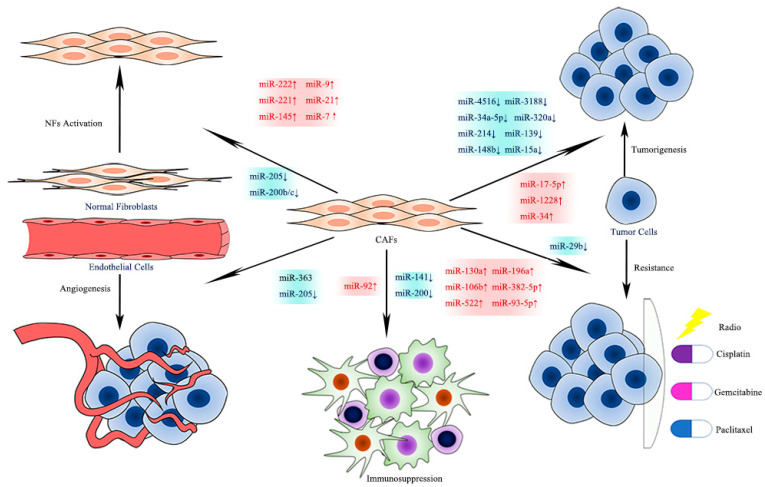
Effects of CAF-derived miRNAs on Tumor Biology.

## Data Availability

The data presented in this study are openly available in PubMed.

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
