# Peer review of "Effects of CAF-Derived MicroRNA on Tumor Biology and Clinical Applications"

_cancers, 2021, doi:10.3390/cancers13133160_

Round 1
Reviewer 1 Report
Summary
In this review, the authors discussed how miRNAs derived from cancer-associated fibroblasts (CAFs) regulate tumorigenesis. The authors described the effects of CAF-derived miRNAs on the feed-forward activation of CAFs, tumor progression in the tumor cells, angiogenesis in the tumor-microenvironment, and chemoresistance in both tumor cells and CAFs. Lastly, the authors discussed the clinical utility of CAF-derived miRNAs as diagnostic/prognostic biomarkers for various human cancers.
Major Comments:
- CAF-derived miRNAs influence multiple processes in the tumor/cancer cells. Although exosome was mentioned in the manuscript to be a carrier of these miRNAs, little is discussed about the mechanism that led to their specific export and targeting of the cell-of-interest. Also, are other miRNA carriers involved in mediating short-/long-distance cell communications? Putting these into perspective will help the readers gain better insights into the life cycle of these CAF-derived miRNAs.
- Although it is fine to adopt a CAF-centric view on tumor biology, we feel that Figure 2 does not present the complexity of crosstalk between various cell types in the tumor microenvironment and how that promotes tumor growth. Also, the mechanism of miRNA-mediated gene silencing was described in the text but wrongly tagged to Figure 2 (Lines 81 to 82). Is there another figure that was left out of the manuscript?
Minor Comments:
- The authors can consider the use of editorial services to reduce grammatical errors. For instance, there is a spelling error at Line 190 – β-cantenin should be β-catenin. We can name many more such typographical and grammatical errors throughout the entire manuscript.
- It will be useful to explain the acronyms used in the manuscript.
- Line 47 – NFs; Line -178 – TNBC; Line 183 – HNC; Line 220 – OSCC
- Line 93 – Table 1 was mentioned in the text but not attached to the manuscript
- References are missing for several important claims
- Line 67 – 70; Lines 91 – 93; Lines 253 – 255; Lines 263 – 266
Author Response
Please see the attachment.
Summary
In this review, the authors discussed how miRNAs derived from cancer-associated fibroblasts (CAFs) regulate tumorigenesis. The authors described the effects of CAF-derived miRNAs on the feed-forward activation of CAFs, tumor progression in the tumor cells, angiogenesis in the tumor-microenvironment, and chemoresistance in both tumor cells and CAFs. Lastly, the authors discussed the clinical utility of CAF-derived miRNAs as diagnostic/prognostic biomarkers for various human cancers.
Response: Thank you for your rigorous consideration.
Major Comments:
- CAF-derived miRNAs influence multiple processes in the tumor/cancer cells. Although exosome was mentioned in the manuscript to be a carrier of these miRNAs, little is discussed about the mechanism that led to their specific export and targeting of the cell-of-interest. Also, are other miRNA carriers involved in mediating short-/long-distance cell communications? Putting these into perspective will help the readers gain better insights into the life cycle of these CAF-derived miRNAs.
Response: We gratefully appreciate for your rigorous consideration and positive comments. Since the subject of our manuscript is about CAF-derived miRNAs, we did not pay more attention to exosomes, which is our negligence. We have added some brief descriptions about exosomes and other carriers of miRNAs in the revised version. Please line 116-126.
Line 116-126: CAF-derived miRNAs regulate the transcriptions of target genes in the nuclei of CAFs or of tumor cells by transporting into CAFs or tumor cells via exosomes derived from CAFs. Exosomes are small secretory vesicles with a diameter pf 40-100nm, and are key determinants to promote cell-to-cell communication 27. Specific miRNA populations are selected for packaging into vesicles, and actively secreted into TME with the assistance of particular molecules. Finally, secreted miRNAs packaged in exosomes are delivered into recipient cells and act the same as endogenous miRNAs to exert gene silencing. In addition, it also suggests that extracellular miRNAs are also actively secreted through a vesicle-free, RNA-binding protein-dependent pathway, which is still not directly evidenced28. In this review, we will not describe the functions of exosomes in detail.
- Although it is fine to adopt a CAF-centric view on tumor biology, we feel that Figure 2 does not present the complexity of crosstalk between various cell types in the tumor microenvironment and how that promotes tumor growth. Also, the mechanism of miRNA-mediated gene silencing was described in the text but wrongly tagged to Figure 2 (Lines 81 to 82). Is there another figure that was left out of the manuscript?
Response: We gratefully appreciate for your rigorous consideration and positive comments. In consideration of the complex functions of various cellular components in TME, we described the functions of different cells in TME in line 43-60, and added a figure (Fig. 1) to highlight the crosstalk between various cell types. The figure “wrongly tagged to Figure 2 (Lines 81 to 82)” was indeed left in the original manuscript, and we have added it as Fig.3 in the revised version.
Line 43-60: The tumor microenvironment (TME), primarily composed of blood and lymphatic vessels, adipose cells, immune cells, fibroblasts, and extracellular matrix (ECM), is vital for tumorigenesis, and the physiological state of TME is closely correlated with each step of tumorigenesis(Fig. 1)1,2. The endothelial cells of blood and lymphatic vessels play a key role in tumor development and immune escaping, which offer nutrients and oxygen and evacuate metabolic wastes and carbon dioxide for tumor growth and development, and help to escape immune surveillance3. Adipose cells, as a major energy source, also contribute to product circulating estrogen, to recruit immune cells and to support angiogenesis1. The immune cells, such as lymphocytes, macrophages and myeloid-derived suppressor cells (MDSCs), are involved in various immune responses orchestrated by tumor. Macrophage within the TME promote angiogenesis, ECM degradation and remodeling, and facilitate tumor cell motility, thus described as “obligate partners for tumor-cell migration, invasion and metastasis”4. MDSCs are effective inhibitor of adaptive immune response to tumors and directly promote metastasis. Meanwhile, tumors are always infiltrated by T regulatory cells (Treg cells), which inhibit adaptive and innate immune in response to tumor stress5. Dysregulation of ECM molecules in cancer progression mediates mechanotransduction as well as tumor initiation and migration.
Minor Comments:
- The authors can consider the use of editorial services to reduce grammatical errors. For instance, there is a spelling error at Line 190 – β-cantenin should be β-catenin. We can name many more such typographical and grammatical errors throughout the entire manuscript.
Response: We apologize for the poor language of our manuscript. We worked on the manuscript for a long time and the repeated addition and removal of sentences and sections obviously led to poor readability. We have now worked on both language and readability. We really hope that the language has been substantially improved.
- It will be useful to explain the acronyms used in the
Line 47 – NFs; Line -178 – TNBC; Line 183 – HNC; Line 220 – OSCC
Response: We gratefully appreciate for your rigorous consideration and suggestions. We have explained the acronyms used in the manuscript. Please see in line 91-92- normal fibroblasts (NFs), line 214- triple-negative breast cancer (TNBC), line 219- head and neck cancer (HNC), line 256-257- oral squamous cell carcinoma (OSCC).
- Line 93 – Table 1 was mentioned in the text but not attached to the manuscript
Response: We gratefully appreciate for your rigorous suggestions. Since the format of Table 1 is not suitable for inserting into the manuscript, and there is no addition information added by Table 1, we decided to delete this table.
- References are missing for several important claims
Line 67 – 70; Lines 91 – 93; Lines 253 – 255; Lines 263 – 266
Response: We apologize for missing the references for the important claims. We have added the citations in the revised manuscript. Please see in line 76-80, line 127-129, line 289-291, line 321-325.
Line 76-80: CAFs are derived from 6 potential original cell types11 (Fig.2): normal fibroblasts12, mesenchymal stem cells (MSCs)13, epithelial cells14 and endothelial cells15, human adipose tissue-derived stem cells (hASCs)16, senescent fibroblasts17, and cancer stem cells (CSCs)18, among which CAFs are principally originated from the transformation of normal fibroblasts and MSCs.
Citations:
- Buchsbaum, R.; Oh, S., Breast Cancer-Associated Fibroblasts: Where We Are and Where We Need to Go. Cancers 2016, 8 (2), 19.
- Ringuette Goulet, C.; Bernard, G.; Tremblay, S.; Chabaud, S.; Bolduc, S.; Pouliot, F., Exosomes Induce Fibroblast Differentiation into Cancer-Associated Fibroblasts through TGF² Signaling. Molecular Cancer Research 2018, 16 (7), 1196-1204.
- Aoto, K.; Ito, K.; Aoki, S., Complex formation between platelet-derived growth factor receptor ² and transforming growth factor ² receptor regulates the differentiation of mesenchymal stem cells into cancer-associated fibroblasts. Oncotarget 2018, 9 (75), 34090-34102.
- Sasaki, K.; Sugai, T.; Ishida, K.; Osakabe, M.; Amano, H.; Kimura, H.; Sakuraba, M.; Kashiwa, K.; Kobayashi, S., Analysis of cancer-associated fibroblasts and the epithelial-mesenchymal transition in cutaneous basal cell carcinoma, squamous cell carcinoma, and malignant melanoma. Hum Pathol 2018, 79, 1-8.
- Lipphardt, M.; Dihazi, H.; Jeon, N. L.; Dadafarin, S.; Ratliff, B. B.; Rowe, D. W.; Müller, G. A.; Goligorsky, M. S., Dickkopf-3 in aberrant endothelial secretome triggers renal fibroblast activation and endothelial mesenchymal transition. Nephrology Dialysis Transplantation 2019, 34 (1), 49-62.
- Jotzu, C.; Alt, E.; Welte, G.; Li, J.; Hennessy, B. T.; Devarajan, E.; Krishnappa, S.; Pinilla, S.; Droll, L.; Song, Y. H., Adipose tissue-derived stem cells differentiate into carcinoma-associated fibroblast-like cells under the influence of tumor-derived factors. Anal Cell Pathol (Amst) 2010, 33 (2), 61-79.
- Alspach, E.; Fu, Y.; Stewart, S. A., Senescence and the pro-tumorigenic stroma. Crit Rev Oncog 2013, 18 (6), 549-58.
- Nair, N.; Calle, A. S.; Zahra, M. H.; Prieto-Vila, M.; Oo, A. K. K.; Hurley, L.; Vaidyanath, A.; Seno, A.; Masuda, J.; Iwasaki, Y.; Tanaka, H.; Kasai, T.; Seno, M., A cancer stem cell model as the point of origin of cancer-associated fibroblasts in tumor microenvironment. Scientific Reports 2017, 7 (1).
Line 127-129: Aberrantly expressed CAF-derived miRNAs, either upregulated or downregulated, appear to have an enormous impact on activation of CAFs, tumorigenesis, metastasis, angiogenesis, immunosuppression, and chemoresistance (Fig.4)7.
Citations:
- Monteran, L.; Erez, N., The Dark Side of Fibroblasts: Cancer-Associated Fibroblasts as Mediators of Immunosuppression in the Tumor Microenvironment. Frontiers in Immunology 2019, 10.
Line 289-291: Tumor angiogenesis is essential for providing adequate nutrition for tumorigenesis and tumor progression52. Anti-angiogenesis therapy is proved effective in various tumor types, which target the cytokine VEGF.
Citations:
- Fukumura, D.; Xavier, R.; Sugiura, T.; Chen, Y.; Park, E. C.; Lu, N.; Selig, M.; Nielsen, G.; Taksir, T.; Jain, R. K.; Seed, B., Tumor induction of VEGF promoter activity in stromal cells. Cell 1998, 94 (6), 715-25.
Line 321-325: Chemoresistance, both acquired and innate, are associated to complex multifactorial processes, including decreased intracellular drug concentrations, hidden drug targets, aberrant regulation of cell survival, and crosstalk between TME and tumor cells23, among which CAFs plays a critical role in the regulation of chemoresistance in various tumor types.
Citations:
- Zhang, T.; Zhang, P.; Li, H. X., CAFs-Derived Exosomal miRNA-130a Confers Cisplatin Resistance of NSCLC Cells Through PUM2-Dependent Packaging. Int J Nanomedicine 2021, 16, 561-577.

Reviewer 2 Report
The Review by Xu Wang et al, entitled: “Effects of CAF-derived MicroRNA on Tumor Biology and Clinical Applications” describes the activities of miRNAs in the crosstalk among tumor cells, endothelial and Cancer-Associated Fibroblasts (CAFs) in the tumor microenvironment (TME).
In the introductory “background” paragraph, the authors provide a concise description of TME and miRNAs.The second paragraph is entitled “CAFs in tumor microenvironment”, and provides some information on CAFs and their origin. Most of the third paragraph entitled: “Introduction of CAF-derived miRNAs” focuses on miRNAs found in tumors and their role in CAF activation, tumor progression, angiogenesis, chemoresistance. The last paragraph on “Clinical applications of CAF-derived miRNA in tumor” relates to miRNAs use as prognostic or diagnostic markers and new therapeutics.
This topic is relevant to Tumor biology, as miRNA are regulators of progression, chemoresistance, neoangiogenesis and are important to prognostics, diagnostics and therapeutics. In this review, the selected literature is updated and refers to the last few years.
However, topics are not well grouped, and the items are not always logically connected, sometimes appearing as a mere listing of papers. The resulting text is not always straightforward and easy to understand. Sometimes there are unjustified repetitions and many typos/grammar mistakes to be corrected.
A few comments follow:
Line 108: “In breast cancer, miR-9, miR-221 and miR-222 were found to be upregulated, while miR-205, miR-200b and miR-200c were downregulated” (no reference provided; in tumor or stromal cells?).
Line 110: Regarding miR-222, its upregulation occurs in CAFs where it targets LBR (abbreviation not reported in extenso) (ref 15).
Line 115: “Significantly higher level of miR-9 in triple-negative breast cancer (TNBC) is involved in the acquisition of a CAF phenotype”: the producing and target cells are not clearly described, and yet, the paragraph title is “CAF-derived miRNAs promoting activation of CAFs”.
Line 119: “These upregulated miRNAs reduce the expression of corresponding signal molecules, and facilitate the activation of CAFs. Inhibitor of this type of miRNAs or analogue of their target molecules will induce a reversion of CAF which may be a novel method of anti-CAF therapeutics.” In general, factors allowing the NAF/CAF conversion may be inhibited to avoid further CAF formation in novel anti-cancer therapies, but the reversion of CAF phenotype is not an obvious consequence and should be demonstrated.
Several suggestions to improve this review are provided:
a.It is suggested that the background paragraph be turned into a description of the TME context from the cellular diversity point of view: a thorough description of peritumoral endothelial, CAFs, immune cells (T cells, macrophages or TAM, MDSC) and their crosstalk is needed. Also, the CAFs paragraph 2 may be fused to 1. Therefore, the description of CAFs would be included in a single first paragraph on TME.
b.In the second paragraph, it is suggested to introduce the miRNA topic, as now covered in paragraph 3, lines 75-95. Then, all the material from line 104 to 161 should be reorganized. One possibility could be grouping based on the miRNAs producing cells (tumor cells versus CAFs). It is also suggested to cover the theme of exosomes, as carriers of miRNAs in TME.In general, because most mechanisms in TME rely on paracrine interactions, attention should be paid to the producing and target cells: if known, this aspect should be always specified.
c.As CAFs are emerging as crucial immune regulators, this topic also deserves attention. This review lacks a description of the effects of CAF-derived miRNA on immune cells (T cells, macrophages etc) in TME.
Suggested papers to be included:
1.CAFs directly inhibit cytotoxic lymphocytes, thus inhibiting killing of tumor cells and driving the recruitment of tumor-infiltrating immune cells toward an immunosuppressive microenvironment. (The Dark Side of Fibroblasts: Cancer-Associated Fibroblasts as Mediators of Immunosuppression in the Tumor Microenvironment. Front Immunol, 2019, 10:1835).
2.A novel mechanism that can induce immune suppression in the tumor microenvironment based on CAF-derived exosomes, generating increased PD-L1 expression and promoting apoptosis and impaired proliferation of T cells. (Dongwei Dou et al., Cancer-Associated Fibroblasts-Derived Exosomes Suppress Immune Cell Function in Breast Cancer via the miR-92/PD-L1 Pathway, Front Immunol, 2020, 11:2026).
3.A miR-141/200a dependent-mechanism causing the accumulation of CXCL12β chemokine in the immunosuppressive CAF-S1 subpopulation (Anne-Marie Givel et al., miR200-regulated CXCL12β promotes fibroblast heterogeneity and immunosuppression in ovarian cancers, Nat Commun 2018 Mar 13;9(1):1056).
Author Response
Please see the attachment.
Comments and Suggestions for Authors
The Review by Xu Wang et al, entitled: “Effects of CAF-derived MicroRNA on Tumor Biology and Clinical Applications” describes the activities of miRNAs in the crosstalk among tumor cells, endothelial and Cancer-Associated Fibroblasts (CAFs) in the tumor microenvironment (TME).
Response: We gratefully appreciate for your rigorous consideration and positive comments.
In the introductory “background” paragraph, the authors provide a concise description of TME and miRNAs.The second paragraph is entitled “CAFs in tumor microenvironment”, and provides some information on CAFs and their origin. Most of the third paragraph entitled: “Introduction of CAF-derived miRNAs” focuses on miRNAs found in tumors and their role in CAF activation, tumor progression, angiogenesis, chemoresistance. The last paragraph on “Clinical applications of CAF-derived miRNA in tumor” relates to miRNAs use as prognostic or diagnostic markers and new therapeutics.
Response: We gratefully appreciate for your rigorous consideration and positive comments.
This topic is relevant to Tumor biology, as miRNA are regulators of progression, chemoresistance, neoangiogenesis and are important to prognostics, diagnostics and therapeutics. In this review, the selected literature is updated and refers to the last few years.
Response: We gratefully appreciate for your rigorous consideration and positive comments.
However, topics are not well grouped, and the items are not always logically connected, sometimes appearing as a mere listing of papers. The resulting text is not always straightforward and easy to understand. Sometimes there are unjustified repetitions and many typos/grammar mistakes to be corrected.
Response: We gratefully appreciate for your rigorous consideration and comments. We apologize for the confusion generated by the previous version of the manuscript and sincerely hope that our logic is now easier to follow with this new version. We have worked on both language and readability. We really hope that the language has been substantially improved.
A few comments follow:
Line 108: “In breast cancer, miR-9, miR-221 and miR-222 were found to be upregulated, while miR-205, miR-200b and miR-200c were downregulated” (no reference provided; in tumor or stromal cells?).
Response: We gratefully appreciate for your rigorous consideration and comments. Maybe we didn’t describe it clearly, causing you to misunderstand. In fact, all the miRNAs we mentioned in the manuscript are derived from CAFs. And we have also added instructions to the revised manuscript. Please see in line 142-144. We apologize for missing the references for the important claims. We have added the citations in the revised manuscript.
Line 142-144: MiR-9, miR-221 and miR-222 derived from CAFs were found to be upregulated, while miR-205, miR-200b and miR-200c were downregulated30-33
Citation:
- Chatterjee, A.; Jana, S.; Chatterjee, S.; Wastall, L. M.; Mandal, G.; Nargis, N.; Roy, H.; Hughes, T. A.; Bhattacharyya, A., MicroRNA-222 reprogrammed cancer-associated fibroblasts enhance growth and metastasis of breast cancer. Br J Cancer 2019, 121 (8), 679-689.
- Baroni, S.; Romero-Cordoba, S.; Plantamura, I.; Dugo, M.; D'Ippolito, E.; Cataldo, A.; Cosentino, G.; Angeloni, V.; Rossini, A.; Daidone, M. G.; Iorio, M. V., Exosome-mediated delivery of miR-9 induces cancer-associated fibroblast-like properties in human breast fibroblasts. Cell Death Dis 2016, 7 (7), e2312.
- Tang, X.; Tu, G.; Yang, G.; Wang, X.; Kang, L.; Yang, L.; Zeng, H.; Wan, X.; Qiao, Y.; Cui, X.; Liu, M.; Hou, Y., Autocrine TGF-² 1/miR-200s/miR-221/DNMT3B regulatory loop maintains CAF status to fuel breast cancer cell proliferation. Cancer Lett 2019, 452, 79-89.
- Du, Y.-E.; Tu, G.; Yang, G.; Li, G.; Yang, D.; Lang, L.; Xi, L.; Sun, K.; Chen, Y.; Shu, K.; Liao, H.; Liu, M.; Hou, Y., MiR-205/YAP1 in Activated Fibroblasts of Breast Tumor Promotes VEGF-independent Angiogenesis through STAT3 Signaling. Theranostics 2017, 7 (16), 3972-3988
Line 110: Regarding miR-222, its upregulation occurs in CAFs where it targets LBR (abbreviation not reported in extenso) (ref 15).
Response: We gratefully appreciate for your rigorous consideration and suggestions. We have explained the acronyms used in the manuscript. Please see in line 145- Lamin B receptor (LBR).
Line 115: “Significantly higher level of miR-9 in triple-negative breast cancer (TNBC) is involved in the acquisition of a CAF phenotype”: the producing and target cells are not clearly described, and yet, the paragraph title is “CAF-derived miRNAs promoting activation of CAFs”.
Response: We gratefully appreciate for your rigorous consideration and comments. Maybe we didn’t describe it clearly, causing you to misunderstand. In fact, all the miRNAs we mentioned in the manuscript are derived from CAFs. In TNBC tissues, miR-9 derived from CAFs is significantly upregulated, which targets the normal fibroblasts and leads to the acquisition of a CAF phenotype and we have also added instructions to the revised manuscript. Please see line 115-117.
Line 115-117: Significantly higher level of CAF-derived miR-9 in triple-negative breast cancer (TNBC) is involved in the acquisition of a CAF phenotype
Line 119: “These upregulated miRNAs reduce the expression of corresponding signal molecules, and facilitate the activation of CAFs. Inhibitor of this type of miRNAs or analogue of their target molecules will induce a reversion of CAF which may be a novel method of anti-CAF therapeutics.” In general, factors allowing the NAF/CAF conversion may be inhibited to avoid further CAF formation in novel anti-cancer therapies, but the reversion of CAF phenotype is not an obvious consequence and should be demonstrated.
Response: We gratefully appreciate for your rigorous consideration and comments. We apologize for imprecise description about the method of anti-CAF therapy. In fact, it has been proved that the reversal of the CAF phenotype is related to the patient's prognosis. We have corrected the statement in the revised version. Please see line 153-157.
Line 153-157: These upregulated miRNAs reduce the expression of corresponding signal molecules, and facilitate the activation of CAFs. Inhibitor of this type of miRNAs or analogue of their tar-get molecules will induce a reversion of CAFs, which may be a novel method to improve prognosis and prolong overall survival, which requires further clinical research to demonstrated.
Several suggestions to improve this review are provided:
a.It is suggested that the background paragraph be turned into a description of the TME context from the cellular diversity point of view: a thorough description of peritumoral endothelial, CAFs, immune cells (T cells, macrophages or TAM, MDSC) and their crosstalk is needed. Also, the CAFs paragraph 2 may be fused to 1. Therefore, the description of CAFs would be included in a single first paragraph on TME.
Response: We gratefully appreciate for your rigorous consideration and comments. In consideration of the complex functions of various cellular components in TME, we described the functions of different cells in TME in line 43-60, and added a figure (Fig. 1) to highlight the crosstalk between various cell types. And we have fused the description of CAFs in paragraph into paragraph 1, please see line 62-80.
Line 43-60: The tumor microenvironment (TME), primarily composed of blood and lymphatic vessels, adipose cells, immune cells, fibroblasts, and extracellular matrix (ECM), is vital for tumorigenesis, and the physiological state of TME is closely correlated with each step of tumorigenesis(Fig. 1)1,2. The endothelial cells of blood and lymphatic vessels play a key role in tumor development and immune escaping, which offer nutrients and oxygen and evacuate metabolic wastes and carbon dioxide for tumor growth and development, and help to escape immune surveillance3. Adipose cells, as a major energy source, also contribute to product circulating estrogen, to recruit immune cells and to support angiogenesis1. The immune cells, such as lymphocytes, macrophages and myeloid-derived suppressor cells (MDSCs), are involved in various immune responses orchestrated by tumor. Macrophage within the TME promote angiogenesis, ECM degradation and remodeling, and facilitate tumor cell motility, thus described as “obligate partners for tumor-cell migration, invasion and metastasis”4. MDSCs are effective inhibitor of adaptive immune response to tumors and directly promote metastasis. Meanwhile, tumors are always infiltrated by T regulatory cells (Treg cells), which inhibit adaptive and innate immune in response to tumor stress5. Dysregulation of ECM molecules in cancer progression mediates mechanotransduction as well as tumor initiation and migration
Line 62-80: Fibroblasts are derived from primitive mesenchyme, with elongated, spindle-like morphology. Normal fibroblasts have a bidirectional effect on tumors: in the early stage of tumorigenesis, fibroblasts maintain structural integrity to most epithelium tissues against the progression of tumor. However, as the malignancy advances, fibroblasts are activated to promote tumor development, which are referred to as cancer-associated fibroblasts (CAFs). CAFs are generally identified by expression of alpha-smooth muscle actin (α-SMA) and fibroblast activation protein (FAP)8. Several other markers can also be used to identify CAFs because of the heterogeneity, including: vimentin, platelet-derived growth factor receptor alpha (PDGFR-α), platelet-derived growth factor receptor beta (PDGFR-β), and fibroblast specific protein (FSP-1)9.
CAFs, consisting of heterogeneous population of mesenchymal cells, exhibit systematic differences in different anatomical sites. Within the same type of tissue, differential expression of CAF markers defines distinct CAF subsets and determines the diversity of CAFs, which may be attributed to the originate cell types, adjacent tissues and occasion of activating10. CAFs are derived from 6 potential original cell types11 (Fig.2): normal fibroblasts12, mesenchymal stem cells (MSCs)13, epithelial cells14 and endothelial cells15, human adipose tissue-derived stem cells (hASCs)16, senescent fibroblasts17, and cancer stem cells (CSCs)18, among which CAFs are principally originated from the transformation of normal fibroblasts and MSCs.
b.In the second paragraph, it is suggested to introduce the miRNA topic, as now covered in paragraph 3, lines 75-95. Then, all the material from line 104 to 161 should be reorganized. One possibility could be grouping based on the miRNAs producing cells (tumor cells versus CAFs). It is also suggested to cover the theme of exosomes, as carriers of miRNAs in TME.In general, because most mechanisms in TME rely on paracrine interactions, attention should be paid to the producing and target cells: if known, this aspect should be always specified.
Response: We gratefully appreciate for your rigorous consideration and comments. We have changed the paragraph 2 into a topic to introduce the miRNAs, please see line 96.
Maybe we didn’t describe it clearly, causing you to misunderstand. In fact, all the miRNAs we mentioned in the manuscript are derived from CAFs. Considering how to organize the material logically, in the first paragraph, we first described the dysregulated MiRNA in breast cancer tissues in the order of upregulation first and then downregulation. In the second paragraph, we described miR-21, which is observed significantly upregulated in different tumor tissues. Through the above two aspects, we proved the role of miRNAs in CAF activation.
In addition, Since the subject of our manuscript is about CAF-derived miRNAs, we did not pay more attention to exosomes, which is our negligence. We have added some brief descriptions about exosomes and other carriers of miRNAs in the revised version. Please line 116-126.
Finally, we gratefully appreciate for your valuable suggestion. We will always pay attention to the producing and target cells of miRNAs.
Line 96: 2. Introduction of CAF-derived miRNAs
Line 116-126: CAF-derived miRNAs regulate the transcriptions of target genes in the nuclei of CAFs or of tumor cells by transporting into CAFs or tumor cells via exosomes derived from CAFs. Exosomes are small secretory vesicles with a diameter pf 40-100nm, and are key determinants to promote cell-to-cell communication 27. Specific miRNA populations are selected for packaging into vesicles, and actively secreted into TME with the assistance of particular molecules. Finally, secreted miRNAs packaged in exosomes are delivered into recipient cells and act the same as endogenous miRNAs to exert gene silencing. In addition, it also suggests that extracellular miRNAs are also actively secreted through a vesicle-free, RNA-binding protein-dependent pathway, which is still not directly evidenced28. In this review, we will not describe the functions of exosomes in detail.
c.As CAFs are emerging as crucial immune regulators, this topic also deserves attention. This review lacks a description of the effects of CAF-derived miRNA on immune cells (T cells, macrophages etc) in TME.
Response: We gratefully appreciate for your rigorous consideration and comments. We have added a paragraph to descript the effects of CAF-derived miRNA on immunosuppression. Please see line 297-318. And we also changed the corresponding figure into Fig. 4.
Line 297-318: 2.4 CAF-derived miRNAs promoting immunosuppression
CAFs were found to enhance the recruitment, differentiation and survival of T regulatory cells (Treg cells), which are generally identified by expression of CD25 and the transcription factor FOXP3, and contributing to generate and maintain an immunosuppressive microenvironment7. According to expression levels of CD29, FAP, FSP1, and SMA, CAFS in High-grade serous ovarian cancers (HGSOC) are distinguished into four subpopulations, CAF-S1, S2, S3 and S4 53. HGSOC are mainly enriched in activated CAF-S1 and CAF-S4 subsets, among which HGSOC enriched in CAF-S1 cells are particularly highly infiltrated by FOXP3+T cells. Compared with CAF-S4 cells, miR-141 and miR-200a are downregulated within CAF-S1 cells, leading to the specific accumulation of CXCL12β,the target of miR-141/200a, in CAF-S1 fibroblasts. Thus, the accumulation of CXCL12β further facilitates the recruitment of CD25+ FOXP3+Treg cells. Treg cells play a key role in fostering immunosuppressive microenvironment, and infiltration of Treg cells predicts shortened overall survival 54.
In addition, miR-92 was found significantly upregulated in CAF-derived exosomes in breast cancer. After transported into breast cells, miR-92 silences its direct target LATS2, which removals the inhibition on YAP1 and promotes the nuclear translocation of YAP1. The occupation of YAP1 in enhancer regions of PD-L1 will increase the transcriptional activity of PD-L1, which promotes T cells tolerance and escapes host immunity 55. In consideration of the function of CAF-miRNAs to promote immunosuppression, the expression levels of miRNAs may be a prospective biomarker to assess the immune status and predict the efficacy of immunotherapy.
Suggested papers to be included:
1.CAFs directly inhibit cytotoxic lymphocytes, thus inhibiting killing of tumor cells and driving the recruitment of tumor-infiltrating immune cells toward an immunosuppressive microenvironment. (The Dark Side of Fibroblasts: Cancer-Associated Fibroblasts as Mediators of Immunosuppression in the Tumor Microenvironment. Front Immunol, 2019, 10:1835).
2.A novel mechanism that can induce immune suppression in the tumor microenvironment based on CAF-derived exosomes, generating increased PD-L1 expression and promoting apoptosis and impaired proliferation of T cells. (Dongwei Dou et al., Cancer-Associated Fibroblasts-Derived Exosomes Suppress Immune Cell Function in Breast Cancer via the miR-92/PD-L1 Pathway, Front Immunol, 2020, 11:2026).
3.A miR-141/200a dependent-mechanism causing the accumulation of CXCL12β chemokine in the immunosuppressive CAF-S1 subpopulation (Anne-Marie Givel et al., miR200-regulated CXCL12β promotes fibroblast heterogeneity and immunosuppression in ovarian cancers, Nat Commun 2018 Mar 13;9(1):1056).
Response: We gratefully appreciate for your introduction to these wonderful research work. According to your suggestion, we properly cite these articles as:
Line 298-318: CAFs were found to enhance the recruitment, differentiation and survival of T regulatory cells (Treg cells), which are generally identified by expression of CD25 and the transcription factor FOXP3, and contributing to generate and maintain an immunosuppressive microenvironment 7. According to expression levels of CD29, FAP, FSP1, and SMA, CAFS in High-grade serous ovarian cancers (HGSOC) are distinguished into four subpopulations, CAF-S1, S2, S3 and S4 53. HGSOC are mainly enriched in activated CAF-S1 and CAF-S4 subsets, among which HGSOC enriched in CAF-S1 cells are particularly highly infiltrated by FOXP3+T cells. Compared with CAF-S4 cells, miR-141 and miR-200a are downregulated within CAF-S1 cells, leading to the specific accumulation of CXCL12β,the target of miR-141/200a, in CAF-S1 fibroblasts. Thus, the accumulation of CXCL12β further facilitates the recruitment of CD25+ FOXP3+Treg cells. Treg cells play a key role in fostering immunosuppressive microenvironment, and infiltration of Treg cells predicts shortened overall survival 54.
In addition, miR-92 was found significantly upregulated in CAF-derived exosomes in breast cancer. After transported into breast cells, miR-92 silences its direct target LATS2, which removals the inhibition on YAP1 and promotes the nuclear translocation of YAP1. The occupation of YAP1 in enhancer regions of PD-L1 will increase the transcriptional activity of PD-L1, which promotes T cells tolerance and escapes host immunity 55. In consideration of the function of CAF-miRNAs to promote immunosuppression, the expression levels of miRNAs may be a prospective biomarker to assess the immune status and predict the efficacy of immunotherapy.
Citation:
- Monteran, L.; Erez, N., The Dark Side of Fibroblasts: Cancer-Associated Fibroblasts as Mediators of Immunosuppression in the Tumor Microenvironment. Frontiers in Immunology 2019, 10.
- Givel, A. M.; Kieffer, Y.; Scholer-Dahirel, A.; Sirven, P.; Cardon, M.; Pelon, F.; Magagna, I.; Gentric, G.; Costa, A.; Bonneau, C.; Mieulet, V.; Vincent-Salomon, A.; Mechta-Grigoriou, F., miR200-regulated CXCL12β promotes fibroblast heterogeneity and immunosuppression in ovarian cancers. Nat Commun 2018, 9 (1), 1056.
55. Dou, D.; Ren, X.; Han, M.; Xu, X.; Ge, X.; Gu, Y.; Wang, X., Cancer-Associated Fibroblasts-Derived Exosomes Suppress Immune Cell Function in Breast Cancer via the miR-92/PD-L1 Pathway. Front Immunol 2020, 11, 2026.

Reviewer 3 Report
In the review, the authors concisely summerized the topic and the review seems basically acceptable for the publication. However, they are advised to add a chapter to discuss the role of tumor-derived extracellular vesicles as a vehicle for miRNA transmission referring to the below article and others.
Oncogene. 2019 Mar;38(12):2162-2176. doi: 10.1038/s41388-018-0564-x.
Author Response
Please see the attachment.
Comments and Suggestions for Authors
In the review, the authors concisely summerized the topic and the review seems basically acceptable for the publication. However, they are advised to add a chapter to discuss the role of tumor-derived extracellular vesicles as a vehicle for miRNA transmission referring to the below article and others.
Response: We gratefully appreciate for your rigorous consideration and positive comments. Since the subject of our manuscript is about CAF-derived miRNAs, we did not pay more attention to exosomes, which is our negligence. We have added some brief descriptions about exosomes and other carriers of miRNAs in the revised version. Please line 116-126.
Line 116-126: CAF-derived miRNAs regulate the transcriptions of target genes in the nuclei of CAFs or of tumor cells by transporting into CAFs or tumor cells via exosomes derived from CAFs. Exosomes are small secretory vesicles with a diameter pf 40-100nm, and are key determinants to promote cell-to-cell communication 27. Specific miRNA populations are selected for packaging into vesicles, and actively secreted into TME with the assistance of particular molecules. Finally, secreted miRNAs packaged in exosomes are delivered into recipient cells and act the same as endogenous miRNAs to exert gene silencing. In addition, it also suggests that extracellular miRNAs are also actively secreted through a vesicle-free, RNA-binding protein-dependent pathway, which is still not directly evidenced28. In this review, we will not describe the functions of exosomes in detail.
Umakoshi M, Takahashi S, Itoh G, Kuriyama S, Sasaki Y, Yanagihara K, Yashiro M, Maeda D, Goto A, Tanaka M.Umakoshi M, et al. Macrophage-mediated transfer of cancer-derived components to stromal cells contributes to establishment of a pro-tumor microenvironment. Oncogene. 2019 Mar;38(12):2162-2176. doi: 10.1038/s41388-018-0564-x.
Response: We gratefully appreciate for your introduction to these wonderful research work. According to your suggestion, we properly cite these articles as:
Line 67-68: CAFs are generally identified by expression of alpha-smooth muscle actin (α-SMA) and fibroblast activation protein (FAP)8.
Citation:
- Umakoshi, M.; Takahashi, S.; Itoh, G.; Kuriyama, S.; Sasaki, Y.; Yanagihara, K.; Yashiro, M.; Maeda, D.; Goto, A.; Tanaka, M., Macrophage-mediated transfer of cancer-derived components to stromal cells contributes to establishment of a pro-tumor microenvironment. Oncogene 2019, 38 (12), 2162-2176.

Round 2
Reviewer 1 Report
The authors made a commendable effort to revise their manuscript based on the reviewers' comments. A few minor comments for their consideration.
- Line 115-117: "CAF-derived miRNAs regulate 115 the transcriptions of target genes in the nuclei of CAFs or of tumor cells by transporting 116 into CAFs or tumor cells via exosomes derived from CAFs." The authors should cite the relevant references.
- Line 356: "FOXA1 promotes the transcription of TGFB3 and then actives the TGF-β signal pathway". Please correct the typographical error.
Author Response
Please see the attachment.
Comments and Suggestions for Authors
The authors made a commendable effort to revise their manuscript based on the reviewers' comments. A few minor comments for their consideration.
Response: We gratefully appreciate for your rigorous consideration and positive comments.
- Line 115-117: "CAF-derived miRNAs regulate the transcriptions of target genes in the nuclei of CAFs or of tumor cells by transporting into CAFs or tumor cells via exosomes derived from CAFs." The authors should cite the relevant references.
Response: We gratefully appreciate for your rigorous consideration and valuable comments. We apologize for missing the reference for the important claim. We have added the citation in the revised manuscript. Please see in line 115-117.
Line 115-117: CAF-derived miRNAs regulate the transcriptions of target genes in the nuclei of CAFs or of tumor cells by transporting into CAFs or tumor cells via exosomes derived from CAFs27.
Citation:
27. Chen, X.; Liang, H.; Zhang, J.; Zen, K.; Zhang, C. Y., Secreted microRNAs: a new form of intercellular communication. Trends Cell Biol 2012, 22 (3), 125-32.
- Line 356: "FOXA1 promotes the transcription of TGFB3 and then actives the TGF-β signal pathway". Please correct the typographical error.
Response: We gratefully appreciate for your rigorous consideration and valuable comments. We apologize for the typographical error. We have corrected it in the revised manuscript. Please see in line 356.
Line 356: FOXA1 promotes the transcription of TGFB3 and then activates the TGF-β signal pathway

Reviewer 2 Report
Regarding the Review entitled: “Effects of CAF-derived MicroRNA on Tumor Biology and Clinical Applications”, the Authors have started to address this Reviewer’s concerns.
Here are some comments to the revised version:
- The titles of paragraphs and figures are still unsatisfying and should be revised by a native English speaker.
- All titles must provide a clear idea of the topic covered in the paragraph or theme depicted in the figure. The first is named: “Background”, the second: “Introduction of CAF-derived miRNAs ”.
3.The topics should be grouped as follows:
- The first paragraph concerning tumor microenvironment (TME) and Cancer-Associated Fibroblasts (CAFs);
- The second paragraph containing general information about the miRNAs characteristics and biosynthesis, together with a description of the relevant/abundant miRNAs found in the TME. The authors should discuss the miRNA localization, consider the many exosomal miRNAs, as well as the autocrine and paracrine mechanisms etc.
-The CAF-derived miRNAs impacting on CAF activation (or NAF/CAF transition) should be taken out from the second paragraph and included in a third paragraph on this specific topic.
The following paragraphs will concern the functional relationship between CAF-derived miRNAs and:
- Tumor progression
- Angiogenesis
-Immunosuppression
-Chemoresistance
-Clinical applications
- Regarding the second paragraph: it has to be noticed that many relevant miRNAs produced by CAFs may be synthesized also by other TME cells (es miR-21), they are not CAF-specific. The review should not disregard this aspect, also considering that it is not always straightforward to assign the synthesis of miRNAs to a specific TME cell type (for example, the exosome-associated miRNAs). “CAF-derived” implies that they originate (i.e.) are synthesized by CAFs.
If the origin is not clear, they cannot be categorized as “CAF-derived”, therefore there are two possibilities.
- Exclude all miRNA not synthesized by CAFs.
- Leave all miRNAs already described in this review but modify the title (for example CAF-associated). This may include miRNAs synthesized by CAFs and exosomes-associated miRNAs uptaken by CAFs but synthesized by other TME cells.
4.I also suggest to list in a Table all miRNAs shown to be synthesized by CAFs, the original tumor type and relative references.
The text is full of grammar and spelling mistakes. There are many sentences with obscure significance, like this: “CAF-derived miRNAs, downregulated in CAFs and exosomes secreted by CAFs, aggravate the progression of tumor via multiple mechanisms, and overexpression of which can restrain the malignancy advance of tumor, which will be a new direction for chemotherapy” (lines 228-231). This review should definitely be re-written by a native English.
Author Response
Comments and Suggestions for Authors
Regarding the Review entitled: “Effects of CAF-derived MicroRNA on Tumor Biology and Clinical Applications”, the Authors have started to address this Reviewer’s concerns.
Response: We gratefully appreciate for your rigorous consideration and positive comments.
Here are some comments to the revised version:
- The titles of paragraphs and figures are still unsatisfying and should be revised by a native English speaker.
Response: We apologize for the poor language of our manuscript. We worked on the manuscript for a long time and the repeated addition and removal of sentences and sections obviously led to poor readability. We have now worked on both language and readability and proofread by English Editing of MDPI. We really hope that the language has been substantially improved. Please see in line 63 and line 84.
Line 63: Figure 1. Prominent components of the TME.
Line 84: Figure 2. Originating cell types of CAFs.
- All titles must provide a clear idea of the topic covered in the paragraph or theme depicted in the figure. The first is named: “Background”, the second: “Introduction of CAF-derived miRNAs”.
Response: We gratefully appreciate for your rigorous consideration and valuable comments. We have changed the titles of the first and second paragraphs. Please see in line 38 and line 84.
Line 38: 1. Tumor microenvironment and cancer-associated fibroblasts
Line 85: 2. Introduction of miRNAs
- The topics should be grouped as follows:
- The first paragraph concerning tumor microenvironment (TME) and Cancer-Associated Fibroblasts (CAFs);
Response: We gratefully appreciate for your rigorous consideration and valuable comments. We have changed the titles of the first paragraph. Please see in line 38.
Line 38: 1. Tumor microenvironment and cancer-associated fibroblasts
- The second paragraph containing general information about the miRNAs characteristics and biosynthesis, together with a description of the relevant/abundant miRNAs found in the TME. The authors should discuss the miRNA localization, consider the many exosomal miRNAs, as well as the autocrine and paracrine mechanisms etc.
Response: We gratefully appreciate for your rigorous consideration and valuable comments. We have changed the titles of the second paragraph. Please see in line 84. And we have added a description of the source and mechanism of secreted miRNA. Please see in line 126-139.
Line 85: 2. Introduction of miRNAs
Line 109-123: miRNAs were previously thought to be unstable molecules, but have been recently demonstrated to circulate in a highly stable and cell-free form in bodily fluids. Circulating miRNAs can be significantly altered in a wide range of pathological conditions, including cancers. The source of such extracellular miRNAs is still not known, but three different pathways have been suggested: 1) passive leakage from broken cells; 2) active secretion from microvesicles, including exosomes and shedding vesicles; and 3) Active secretion using a microvesicle-free, RNA-binding protein-dependent pathway, such as high-density lipoprotein (HDL)27. Exosomes are small secretory vesicles with a diameter of 40-100nm, which are key determinants promoting cell-to-cell communication 28. Specific miRNA populations are selected for packaging into vesicles, and actively secreted into the TME with the assistance of particular molecules. Finally, secreted miRNAs packaged in exosomes are delivered into recipient cells, and act the same as endogenous miRNAs to exert gene silencing. Similarly, HDL can readily associate with exogenous miRNAs and deliver them to recipient cells 27.
-The CAF-derived miRNAs impacting on CAF activation (or NAF/CAF transition) should be taken out from the second paragraph and included in a third paragraph on this specific topic.
Response: We gratefully appreciate for your rigorous consideration and valuable comments. We have adjusted the structure of the article, and we put the effects of CAF-derived miRNA on CAF activation, tumor progression, angiogenesis, immunosuppression and chemoresistance in the third paragraph, and the clinical application in the fourth paragraph. We think that the description of miRNA biological functions in the third paragraph and the clinical application of miRNA in the fourth paragraph will make the review more logical.
The following paragraphs will concern the functional relationship between CAF-derived miRNAs and:
- Tumor progression
- Angiogenesis
-Immunosuppression
-Chemoresistance
-Clinical applications
Response: We gratefully appreciate for your rigorous consideration and valuable comments. We have adjusted the structure of the article, and we put the effects of CAF-derived miRNA on CAF activation, tumor progression, angiogenesis, immunosuppression and chemoresistance in the third paragraph, and the clinical application in the fourth paragraph. We think that the description of miRNA biological functions in the third paragraph and the clinical application of miRNA in the fourth paragraph will make the review more logical.
- Regarding the second paragraph: it has to be noticed that many relevant miRNAs produced by CAFs may be synthesized also by other TME cells (es miR-21), they are not CAF-specific. The review should not disregard this aspect, also considering that it is not always straightforward to assign the synthesis of miRNAs to a specific TME cell type (for example, the exosome-associated miRNAs). “CAF-derived” implies that they originate (i.e.) are synthesized by CAFs.
If the origin is not clear, they cannot be categorized as “CAF-derived”, therefore there are two possibilities.
- Exclude all miRNA not synthesized by CAFs.
2) Leave all miRNAs already described in this review but modify the title (for example CAF-associated). This may include miRNAs synthesized by CAFs and exosomes-associated miRNAs uptaken by CAFs but synthesized by other TME cells.
Response: We gratefully appreciate for your rigorous consideration and valuable comments. Maybe we didn’t describe it clearly, causing you to misunderstand. In fact, all the miRNAs we mentioned in the manuscript are derived from CAFs. For example, although the miR-21 was highly expression in both CAFs and lung adenocarcinoma cells, no difference in survival was detected between lung adenocarcinoma cases with high level or low level miR-21. In contrast, a significantly shorter overall survival was observed in patients with lung adenocarcinoma with upregulated miR-21CAFs compared to those with downregulated miR-21 CAFs. Meanwhile, in PDAC, the expression of miR-21 in tumor cells did not change much with gemcitabine treatment, while in CAFs, the gemcitabine-treated cells showed about four times the amount of miR-21 expression as the control group cells. In addition, in the study of miR-21-mediated metabolic alteration of CAFs, the expression level of miR-21 was significantly higher in pancreatic CAFs and that the co-culture of these CAFs and pancreatic cancer cell lines accelerated tumor progression. Similarly, all the studies included in this review focus on miRNAs that are differentially expressed in CAFs and play a role in tumor progression, since we have excluded functional miRNAs derived from other cell types in the process of collecting literature. Therefore, in this review, what we are focusing on is the differentially expressed miRNAs in CAF, rather than in tumor cells or other TME cells in tumor tissues, so we believe that our use of “CAF-derived” is reasonable. As for the expression of “CAF-associated”, we believe that this will expand the scope of our review, and will lead to the need for our review to supplement the miRNAs from a variety of cells that have effects on CAFs. Obviously, this cannot be done in 10 days, so we don't think such a change should be made.
The references for the description of miR-21 above are as follows:
- Kunita, A.; Morita, S.; Irisa, T. U.; Goto, A.; Niki, T.; Takai, D.; Nakajima, J.; Fukayama, M., MicroRNA-21 in cancer-associated fibroblasts supports lung adenocarcinoma progression. Sci Rep 2018, 8 (1), 8838.
- Zhang, L.; Yao, J.; Li, W.; Zhang, C., Micro-RNA-21 Regulates Cancer-Associated Fibroblast-Mediated Drug Resistance in Pancreatic Cancer. Oncol Res 2018, 26 (6), 827-835.
- Chen, S.; Chen, X.; Shan, T.; Ma, J.; Lin, W.; Li, W.; Kang, Y., MiR-21-mediated Metabolic Alteration of Cancer-associated Fibroblasts and Its Effect on Pancreatic Cancer Cell Behavior. Int J Biol Sci 2018, 14 (1), 100-110.
- I also suggest to list in a Table all miRNAs shown to be synthesized by CAFs, the original tumor type and relative references.
Response: We gratefully appreciate for your rigorous suggestions. Since the format of Table 1 is not suitable for inserting into the manuscript, and there is no addition information added by Table 1, we decided to delete this table. However, in order to satisfy your opinion, we will upload the form as an attachment.
The text is full of grammar and spelling mistakes. There are many sentences with obscure significance, like this: “CAF-derived miRNAs, downregulated in CAFs and exosomes secreted by CAFs, aggravate the progression of tumor via multiple mechanisms, and overexpression of which can restrain the malignancy advance of tumor, which will be a new direction for chemotherapy” (lines 228-231). This review should definitely be re-written by a native English.
Response: We gratefully appreciate for your rigorous consideration and comments. We apologize for the confusion generated by the previous version of the manuscript and sincerely hope that our logic is now easier to follow with this new version. We have worked on both language and readability and proofread by English Editing of MDPI. We really hope that the language has been substantially improved.
Line 238-241: CAF-derived miRNAs, downregulated in CAFs and exosomes secreted by CAFs, aggravate tumor progression through multiple mechanisms, the overexpression of which can restrain the malignancy advance, thus providing a new direction for chemotherapy.

Reviewer 3 Report
The manuscript seems to have revised responding to reviewers' comments and to attain sufficient quality for the publication.
Author Response
Comments and Suggestions for Authors
The manuscript seems to have revised responding to reviewers' comments and to attain sufficient quality for the publication
Response: We gratefully appreciate for your rigorous consideration and positive comments.

Round 3
Reviewer 2 Report
Most of this reviewer's concerns have been addressed. Still, there are some changes to be introduced, as follows:
Line 84: Figure 2. Originating cell types of CAFs. Change in "ORIGIN OF CAFs"
Figure 3. Production of miRNAs. Change in "PROCESSING OF miRNAs"
Line 85: 2. Change to "INTRODUCTION TO miRNAs"
Line 109-123: Change to: "Finally, secreted miRNAs packaged in exosomes are delivered into recipient cells, and act similarly to endogenous miRNAs.."
Title of paragraph 4, change to: 4. Clinical applications of CAF-derived miRNAs in tumors.
Tile of paragraph 5, change to: " 5. Conclusions and future perspectives"
Please read and include the following important reference: “A framework for advancing our understanding of cancer-associated fibroblasts”, by Sahai et al., Nature Reviews Cancer volume 20, pages 174–186, 2020.
Author Response
COMMENTS TO THE AUTHOR:
Reviewer #2
Comments and Suggestions for Authors
Most of this reviewer's concerns have been addressed. Still, there are some changes to be introduced, as follows:
Response: We gratefully appreciate for your rigorous consideration and positive comments.
Line 84: Figure 2. Originating cell types of CAFs. Change in "ORIGIN OF CAFs"
Response: We gratefully appreciate for your rigorous consideration and valuable comments. We have changed the titles of Figure 2. Please see in line 84.
Line 84: Figure 2. Origin of CAFs.
Figure 3. Production of miRNAs. Change in "PROCESSING OF miRNAs"
Response: We gratefully appreciate for your rigorous consideration and valuable comments. We have changed the titles of Figure 3. Please see in line 107.
Line 107: Figure 3. Processing of miRNAs.
Line 85: 2. Change to "INTRODUCTION TO miRNAs"
Response: We gratefully appreciate for your rigorous consideration and valuable comments. We have changed the titles of paragraph 2. Please see in line 85.
Line 85: 2. Introduction to miRNAs
Line 109-123: Change to: "Finally, secreted miRNAs packaged in exosomes are delivered into recipient cells, and act similarly to endogenous miRNAs.."
Response: We gratefully appreciate for your rigorous consideration and valuable comments. We have made changes in line 119-123. Please see in line 119-123.
Line 119-123: Finally, secreted miRNAs packaged in exosomes are delivered into recipient cells, and act similarly to endogenous miRNAs to exert gene silencing.
Title of paragraph 4, change to: 4. Clinical applications of CAF-derived miRNAs in tumors.
Response: We gratefully appreciate for your rigorous consideration and valuable comments. We have changed the titles of paragraph 4. Please see in line 380.
Line 380: 4. Clinical applications of CAF-derived miRNAs in tumors
Tile of paragraph 5, change to: " 5. Conclusions and future perspectives"
Response: We gratefully appreciate for your rigorous consideration and valuable comments. We have changed the titles of paragraph 5. Please see in line 414.
Line 414: 5. Conclusions and future perspectives
Please read and include the following important reference: “A framework for advancing our understanding of cancer-associated fibroblasts”, by Sahai et al., Nature Reviews Cancer volume 20, pages 174–186, 2020.
Response: We gratefully appreciate for your introduction to these wonderful research work. We believe that the identification and origin of CAFs in this review are applicable to our manuscript, so we properly cite this review as:
Line 69-70: CAFs are generally identified by the expression of alpha-smooth muscle actin (α-SMA) and fibroblast activation protein (FAP)8, 9.
Line 78-82: CAFs are derived from six potential original cell types12 9(Fig. 2): normal fibroblasts13, mesenchymal stem cells (MSCs)14, epithelial cells15 and endothelial cells16, human adipose tissue-derived stem cells (hASCs)17, senescent fibroblasts18, and cancer stem cells (CSCs)19, among which CAFs principally originate from the transformation of normal fibroblasts and MSCs.
Citation:
- Sahai, E.; Astsaturov, I.; Cukierman, E.; DeNardo, D. G.; Egeblad, M.; Evans, R. M.; Fearon, D.; Greten, F. R.; Hingorani, S. R.; Hunter, T., et al., A framework for advancing our understanding of cancer-associated fibroblasts. Nat Rev Cancer 2020, 20 (3), 174-186.
